# Corrosion Behavior of Sensitized AISI 304 Stainless Steel in Acid Chloride Solution

**DOI:** 10.3390/ma15238543

**Published:** 2022-11-30

**Authors:** Viera Zatkalíková, Milan Uhríčik, Lenka Markovičová, Lenka Kuchariková

**Affiliations:** Department of Materials Engineering, Faculty of Mechanical Engineering, University of Žilina, Univerzitná 8215/1, 010 26 Žilina, Slovakia

**Keywords:** sensitization, austenitic stainless steel, corrosion, chromium depletion, potentiodynamic polarization, exposure immersion test

## Abstract

Corrosion resistance of sensitized austenitic stainless steel (SS) in chloride environments is currently the subject of numerous studies. Most of them are focused on neutral chloride solutions at room temperature and the experiments are carried out on ground stainless steels surfaces. This paper deals with the corrosion behavior of sensitized AISI 304 stainless steel in acid 1 M chloride solution (pH = 1.1) at the temperatures of 20 ± 3 °C and 50 °C. The specimens after sensitization are tested as covered by high-temperature surface oxides (“heat tinted”), and also after their chemical removal to assess the impact of the surface state on corrosion resistance. Potentiodynamic polarization (PP) and exposure immersion test are used as the independent corrosion tests. Microstructure before/after exposure immersion test is evaluated by optical microscopy (OM) and SEM. The results obtained showed that sensitization significantly conditions corrosion regardless of the removal of high-temperature oxides, and the elevated temperature mainly acts as its accelerating factor.

## 1. Introduction

AISI 304 is a widely used austenitic stainless steel (SS) recommended for applications that require a combination of high corrosion resistance, high strength, ductility and malleability, weldability, non-magnetic behavior and low cost. The protective passive film on the SS surface ensures high resistance to the uniform corrosion in common oxidation environments, but under special conditions local corrosion forms can be initiated [1,2]. The presence of aggressive substances can evoke local breakdown of passivity and dangerous pitting corrosion [3,4,5,6,7]. An exposure of austenitic SS in the temperature range of 500–800 °C (“critical temperatures”) with consequent slow cooling in the air, e.g., during welding, is related to their susceptibility to another form of local corrosion—intergranular. The precipitation of chromium carbides in critical temperatures consumes chromium from a narrow band along the grain boundary and this makes the zone anodic to the unaffected grains. The chromium content drop under the passivity limit (11.5 wt %) near the grain boundaries leads to the sensitization of this material, which becomes susceptible to intergranular corrosion in aggressive environments [8,9,10,11,12]. Moreover, chromium depleted zones perform the preferential path for other local corrosion attacks and for crack propagation if under tensile stress [13,14,15].

In addition to sensitization, heating of stainless steels at the critical temperatures (e.g., during welding) is accompanied by the formation of colored high-temperature oxide film (heat tint) caused by the oxidation of chromium during a thermal process [16,17,18]. Regions below the heat tint become chromium depleted and susceptible to local corrosion [16]. The authors of [17] evaluated the corrosion resistance of heat tinted (200–1050 °C) AISI 304 SS specimens by potentiodynamic polarization (PP) in chloride solutions (0.02–2.2 wt % Cl^−^). Depending on the temperature of heat treatment, they observed the heat tint color variation from yellowish to the dark blue/black related to the thickness of the surface oxides layer. For all tested Cl^−^ concentrations they recorded a strong decrease of the pitting potential for 600 °C heat tinted specimens compared to the 200 °C ones. Because of the negative effect on the local corrosion resistance, the authors of [16,17,18] recommend the heat tint removal. In order to review corrosion resistance, Mahajanam et al. [16] compared PP curves of sensitized AISI 316 SS specimens after various cleaning treatments (two kinds of mechanical treatment and three kinds of chemical treatment). The combination of pickling and passivation in the mixture of hydrochloric and nitric acids in water solution was found to be the best option for heat tint removal. The same chemical treatment is also recommended by the authors of [17,18].

Currently, the relation between sensitization and the pitting corrosion susceptibility of austenitic SSs is the subject of numerous studies [13,14,19,20,21,22,23]. In addition to the effect of chromium-depleted zones, the authors of [14,20,21] presented the role of MnS inclusions on the pitting corrosion initiation of sensitized austenitic SSs. According to the available references, most electrochemical PP tests were carried out on the ground surface in pH neutral chloride solutions at room temperature [14,19,20,21]. The authors of [22] compared the effect of various mechanical surface treatments on the pitting corrosion resistance of sensitized AISI 304 SS in 3.5 wt % solution at 23 and 50 °C. Because the studies that evaluate the corrosion behavior of sensitized austenitic SSs at elevated temperatures in low pH chloride solutions are lacking, our article is focused on these conditions.

The objective of this study is the evaluation of the corrosion resistance of sensitized AISI 304 stainless steel in acid 1 M chloride solution. To assess the impact on corrosion resistance, the specimens after sensitization are evaluated both as heat tinted and after chemical removal of high temperature oxides. Potentiodynamic polarization and exposure immersion test, carried out at both room (20 ± 3 °C) and elevated (50 °C) temperatures, are used as the independent corrosion tests. Microstructure of as received and sensitized experimental material before/after exposure immersion test is evaluated by optical microscopy (OM) and SEM.

## 2. Materials and Methods

The material used was AISI 304 austenitic SS in sheets of 1.5 mm thickness with 2B surface finish (smooth and matte metallic glossy surface). The chemical composition obtained by X-ray fluorescence is listed in Table 1.

Microstructure of as received AISI 304 SS (Figure 1) observed by OM is polyhedral, austenitic with numerous twins that could be related to annealing or rolling. According to the chemical composition of SS, visible inclusions could contain Mn [14,20,21].

The rectangular specimens (15 mm × 40 mm × 1.5 mm) were prepared for an “improper” heat treatment to evoke a sensitization and for consequent corrosion tests. A part of the specimens was left without heat treatment (in the as received state) for a comparison of the corrosion test results.

The heat exposure of specimens was performed in a furnace at 650 °C for 40 h with consequent slow cooling in the air to create suitable diffusion conditions for precipitation of chromium carbides (conditions were chosen according to the diagram of carbon solubility in austenite [24]). Sensitization of heat treated specimens was confirmed by the oxalic acid etching test performed according to A practice of ASTM A262 standard method [25] under the conditions listed in Table 2. Before the etching test the specimen surface was prepared metallographically, rinsed with ethanol and air-dried. During electrochemical etching, the specimen was connected to the positive pole as the anode (+); the cathode (−) was SS block. The cathode and anode were mutually parallel (distance 5 mm). The etched surface was evaluated using OM and SEM [25].

Before the corrosion tests, half of the sensitized specimens were chemically treated by pickling (conditions in Table 3) for removal the high-temperature oxides. The other specimens were left as heat tinted. The overview of specimen types with their designations is given in Table 4.

The 1 M chloride solution was used as the basic corrosion environment for both performed corrosion tests. This solution was represented by 5 wt % FeCl_3_ (pH = 1.1, redox potential 0.691 V) for exposure immersion test (modified version of ASTM G48 standard) and by 0.9 M NaCl + 0.1 M HCl (pH = 1.1) with the same chloride concentration but with lower aggressiveness (redox potential 0.509) for potentiodynamic polarization (FeCl_3_ solution is too aggressive for our corrosion cell). All chemical compounds used in experiments were analytical grade.

Both corrosion tests were carried out at the temperatures 20 ± 3 °C and 50 °C.

The potentiodynamic polarization was performed in the conventional three-electrode cell system with a calomel reference electrode (SCE, +0.248 V vs. SHE at 20 °C) and a platinum auxiliary electrode (Pt) using BioLogic corrosion measuring system with PGZ 100 measuring unit (BioLogic, Seyssinet-Pariset, France). The time for potential stabilization between the specimen and the electrolyte was set to 10 min. The exposed area of a specimen was 1 cm^2^.

The potentiodynamic polarization curves were recorded at the sweep rate of 1 mV/s [26,27], a potential scan range was applied between −0.3 and 0.9 V vs. open circuit potential (OCP). At least three experiment repeats were carried out for each specimen and the representative curve was selected.

The specimen shape for 24 h exposure immersion test was rectangular (15 mm × 40 mm × 1.5 mm). The specimens were degreased by ethanol and weighed out with accuracy ± 0.00001 g before the test. The group of three parallel specimens was tested for each type of surface. After exposure, the specimens were brushed, washed by demineralized water, freely dried and weighted out again [28]. After exposure, the corrosion resistance was evaluated by calculated corrosion rates (g/(m^2^ h)). The pitted specimen surfaces were observed and assessed by OM and SEM.

## 3. Results and Discussion

### 3.1. Verification of Sensitization by Oxalic Acid Test

The microstructure of the heat exposed specimen (650 °C/40 h) after the oxalic acid etching test is shown in Figure 2a,b. According to ASTM A262 practice A standard the observed etch microstructure can be considered the ditch one because the grain boundaries are completely surrounded by ditches that arose by carbide dissolution [8,25]. This result confirms sensitization obtained during performed heat exposure. For comparison, the etch microstructure of the as received specimen (Figure 2c) appeared to be stepped, which can be caused by the different dissolution rates of the variously oriented grains and it is not related to the chromium carbides precipitation [8,9,10,25].

### 3.2. Potentiodynamic Polarization

The PP curves of tested specimens measured at 20 ± 3 °C are shown in Figure 3, the PP curves for 50 °C are presented in Figure 4. Values of the electrochemical PP parameters are listed in Table 5.

The polarization curves with the passive anodic branches typical for passivating metals (as received 20, SP 20 and as received 50) are evaluated by the pitting potentials E_p_ [19,20,21,29] and by the corrosion potentials E_corr_ determined directly from PP curves. E_p_ values were determined as the potentials of strong permanent increase of the current density in the passivity region, which indicates a breakdown of the passive film and the onset of the stable pit growth. The higher E_p_ value means the higher resistance to the pitting corrosion [3,4,6,19,20,22]. E_corr_ values were determined as the potentials of the transition from the cathodic to the anodic branches. A shift of E_corr_ in the positive direction points to a higher thermodynamic stability of the material.

The other curves (S 20, S 50 and SP 50) do not show the passive behavior and it points to the active anodic dissolution. These curves are characterized by the corrosion potentials E_corr_ and by the corrosion current densities i_corr_ obtained by Tafel extrapolation using EC-LAB software that generated the E_corr_ and i_corr_ values. Corrosion current density i_corr_ expresses the kinetics of corrosion reactions. The higher i_corr_ means the higher corrosion rate [26].

As can be assessed from PP curves and from the PP parameters values, the sensitization caused the marked decrease of the corrosion resistance at the both temperatures. Chromium depletion along the sensitized grain boundaries (Figure 2a,b) could cause a decrease of the passive film homogeneity and stability [13,14,21]. It led to the loss of the passivity and to the active anodic dissolution in aggressive acid Cl^−^ solution. The SP 20 curve (in Figure 3) can be considered an exception from the above mentioned behavior. As can be seen, this curve has passive branch but E_p_ value (−0.05 V vs. SCE) is markedly lower compared to the as received 20 curve (0.22 V vs. SCE). For this specimen, the positive effect of the high-temperature oxides chemical removal was registered [16,17]. Nitric acid as a part of the pickling solution could strengthen the passive surface film [30,31,32], which partially retained its protectiveness. This was reflected in the curve with a narrow passivity region. The authors [30,31,32] recorded that nitric acid passivation contributes to the changes in the surface chemistry by oxidation of chromium and dissolution of iron oxides. The result is an increase in the Cr/Fe ratio.

The authors [19,20,21] dealt with the pitting corrosion resistance of sensitized AISI 304 SS with ground surface in pH neutral chloride solutions. By potentiodynamic polarization at the room temperature, they all obtained PP curves with passive anodic branches but with decreased pitting potentials compared to the state without sensitization. Cheng et al. [19] recorded E_p_ = 0.25 V vs. SSE for sensitized AISI 304 in 0.1 M MgCl_2_ solution. The similar E_p_ value (0.3 V vs. SSE) in the same solution was obtained by Tokuda et al. [20]. Hou et al. [21] used 3.5 wt % NaCl solution and measured a higher E_p_ value (0.4 V vs. SSE). Taking into account the SSE reference potential 0.654 V (i.e., +0.413 V vs. SCE), the E_p_ of SP 20 specimen (0.36 V vs. SSE) is close to the value of the authors [21] but under different conditions (surface state, pH).

According to the obtained results (Figure 4, Table 5), the temperature of 50 °C was manifested as a significant factor reducing the corrosion resistance. It was also documented by the authors of [22,28,33,34,35,36]. Under this temperature conditions chemical surface treatment for removal of the high-temperature oxides (Figure 4, SP 50 specimen) did not prevent the passivity lose and the active dissolution with high corrosion current density took place. This could be related to the temperature changes in hydrolysis kinetics, increase of the diffusion rate and the stronger chemisorption of chloride anions and their consequent intensive penetration inside the material [22]. The authors of [37,38] explained that an increase of electrolyte temperature gradually weakens the self-repairing ability of SS passive film, and the destruction rate is much higher than the self-repairing one. It leads to the breakdown of the equilibrium stage and the system reaches an accelerated disruption state with sharp increase of the current density. Ezuber et al. [22] performed PP measurements on sensitized AISI 304 SS (after 24 h sensitization) at the same temperature but differently in pH neutral 3.5 wt % NaCl solution and on the ground surface. They observed marked E_p_ decrease compared to the room temperature (E_p_ = −0.095 V vs. SCE at 50 °C; E_p_ = 0.15 V vs. SCE at 23 °C) but not a loss of the passivity.

### 3.3. Exposure Immersion Test

As can be seen from Figure 5, the sensitized AISI 304 SS specimens were attacked by the pitting corrosion visible to the naked eye under the conditions of 24-h exposure (20 ± 3 °C/50 °C, 1 M Cl^−^ acid solution) regardless of the previous chemical removal of the high-temperature oxides. A similar round shape of the pits was noted in the heat-affected zones of welded AISI 304 SS after exposure in the same solution by the authors [14].

The average corrosion rates calculated from the mass losses of the specimens (mass loss per unit area per unit time, g/(m^2^ h)) are presented in Table 6.

The OM micrographs of cross sections (Figure 6 and Figure 7) capturing the edges of the pits point to the close relation between sensitization and the pitting. Regardless of the solution temperature and the state of the surface (presence/absence of high temperature oxides), chromium-depleted zones adjacent to the grain boundaries obviously became the sites of pits nucleation. This phenomenon is clearly visible also in the SEM micrograph of the surface area after oxalic acid electroetching (Figure 8). The similar initiation of the pitting, for the same sensitized SS was also recorded by the authors of [13,19] and for the AISI 403 SS by the authors of [39]. According to the Figure 6 and Figure 7 the preferential initiation of corrosion at MnS inclusions described by the authors of [14,20,21] does not seem likely.

Unlike PP, the exposure test did not show a marked difference in corrosion resistance between the S and SP specimens at the temperature of 20 ± 3 °C. It could be related to the higher aggressiveness of 5 wt % FeCl_3_ solution (redox potential 0.691 V) used as 1 M Cl^−^ solution for exposure test compared to 0.9 M NaCl + 0.1 M HCl (redox potential 0.509 V) used for PP. Differences in corrosion process during electrochemical and exposure immersion tests could also have contributed to the observed phenomenon [40,41]. In potentiodynamic polarization, the corrosion process is influenced by a controlled change of potential in the anodic direction from minimum to maximum with a selected sweep rate (mV/s). The obtained results also depend on the time of potential stabilization between the specimen and the electrolyte before the test and on the set sweep rate. Unlike the PP test, in the exposure immersion test, the potential between the specimen and the electrolyte changes naturally depending on the ongoing oxidation and reduction reactions. It is also affected by the diffusion of reaction components, especially chloride anions from the surrounding environment. In addition, during the phase of stable growth of corrosion pits, the pH decreases due to the hydrolysis (Fe^2+^ + 2H_2_O → Fe(OH)_2_ + 2H^+^) [14,41].

According to the Figure 6 and Figure 7, there is not a marked difference in appearance of pit edges dependent on the considered temperatures (20 ± 3 and 50 °C). Higher temperature could result in acceleration of diffusion [22] and intensive penetration of chloride anions through the weakened sites of the passive film. It was manifested by significant increase in corrosion rates (Table 6).

## 4. Conclusions

The heat exposure at 650 °C/40 h with slow cooling in air evoked sensitization of AISI 304 SS confirmed by oxalic acid electroetching test.Both independent corrosion tests showed decrease in corrosion resistance in acid chloride solution after sensitization.PP carried out at 20 ± 3 °C revealed a difference in corrosion behavior between the heat tinted (S) and chemically treated (SP) specimens. PP curve for SP specimen showed a narrow passivity region (E_p_ = −0.05 ± 0.04 V vs. SCE) the S specimen curve reflected active anodic dissolution. According to the PP curves at 50 °C, both S and SP specimens lost their passive behavior.24 h exposure in acid chloride solution evoked the pitting corrosion of all sensitized specimens.OM micrographs of the cross sections (Figure 6 and Figure 7) revealed the close relationship between the sensitization and the pitting—chromium depleted localities near grain boundaries became the sites of the pit nucleation.The performed exposure test did not confirm a higher corrosion resistance of SP specimens compared to S specimens at 20 °C.According to the average corrosion rates (Table 6) the temperature of 50 °C significantly affected the corrosion kinetics. This correlates with results of PP.

On the basis of the performed experiments, it can be concluded that several hours’ exposure of sensitized AISI 304 SS in the acid 1 M chloride solution (pH 1.1) can start pitting corrosion regardless the removal of high-temperature oxides. At the temperature of 50 °C, the corrosion process is significantly accelerated.

## Figures and Tables

**Figure 1 materials-15-08543-f001:**
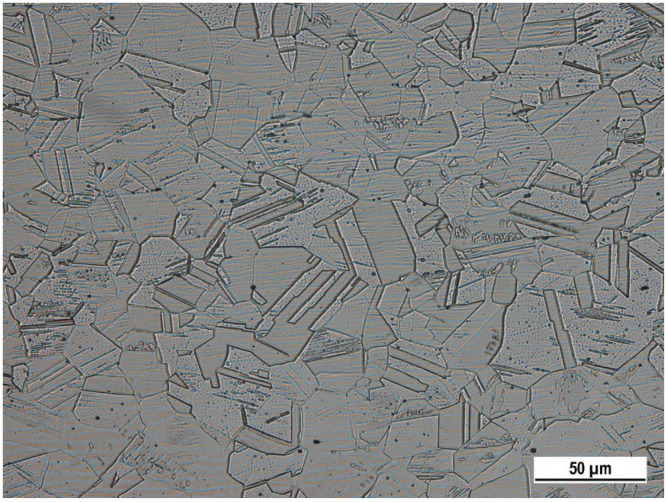
Microstructure of AISI 304 SS, longitudinal section (glycerine + HNO_3_ + HCl etch., OM, magnification 500×).

**Figure 2 materials-15-08543-f002:**
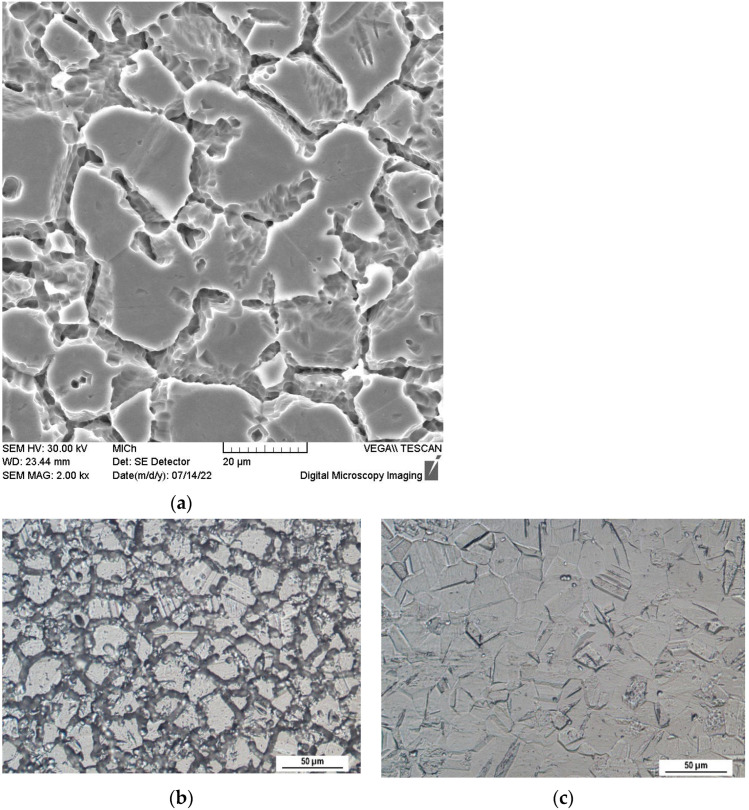
Microstructure of AISI 304 SS after oxalic acid electroetching according ASTM A262 practice A: (**a**) after heat exposure, SEM, magnification 2000×; (**b**) after heat exposure, OM, magnification 400×; (**c**) as received, OM, magnification 400×.

**Figure 3 materials-15-08543-f003:**
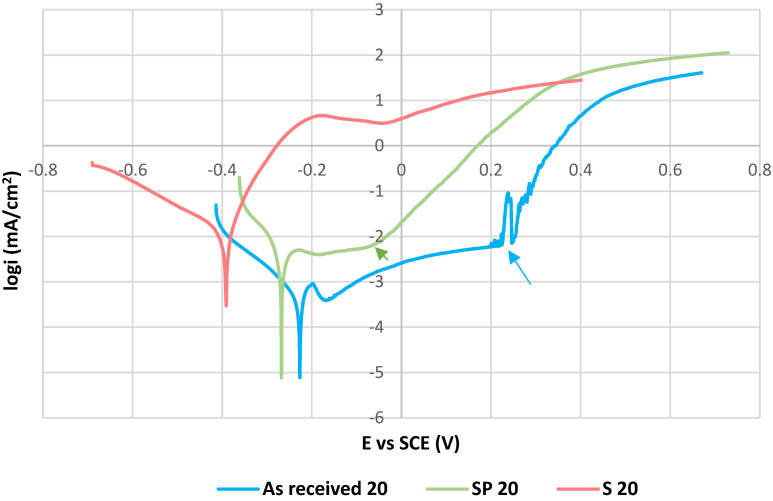
Potentiodynamic polarization curves for tested specimens at 20 ± 3 °C. The pitting potentials E_p_ related to the as received 20 and SP 20 curves are marked by the arrows.

**Figure 4 materials-15-08543-f004:**
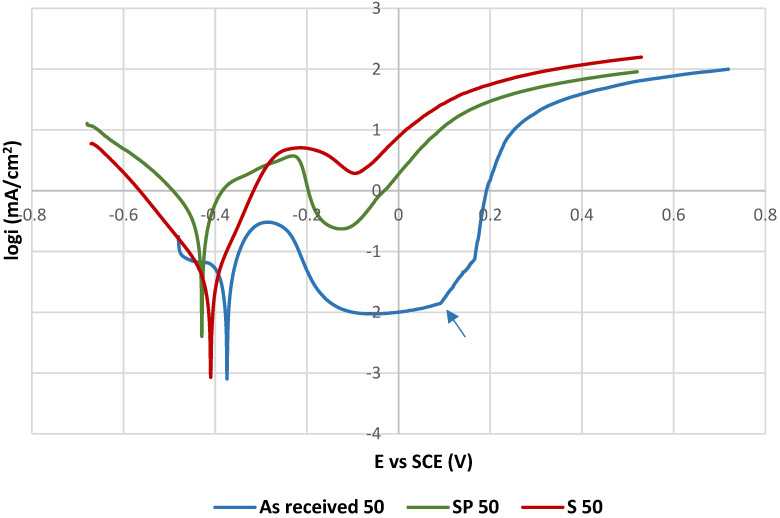
Potentiodynamic polarization curves for tested specimens at 50 °C. The pitting potential E_p_ related to the as received 50 curve is marked by the arrow.

**Figure 5 materials-15-08543-f005:**
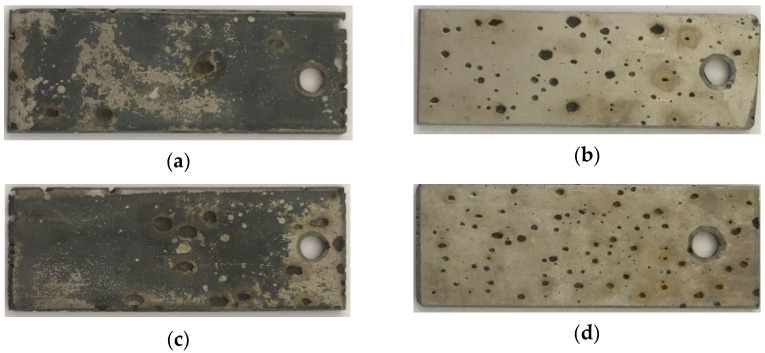
The sensitizied specimens after 24 h exposure in 1 M Cl^−^ solution (represented by 5 wt % FeCl_3_ solution): (**a**) S 20; (**b**) SP 20; (**c**) S 50; (**d**) SP 50.

**Figure 6 materials-15-08543-f006:**
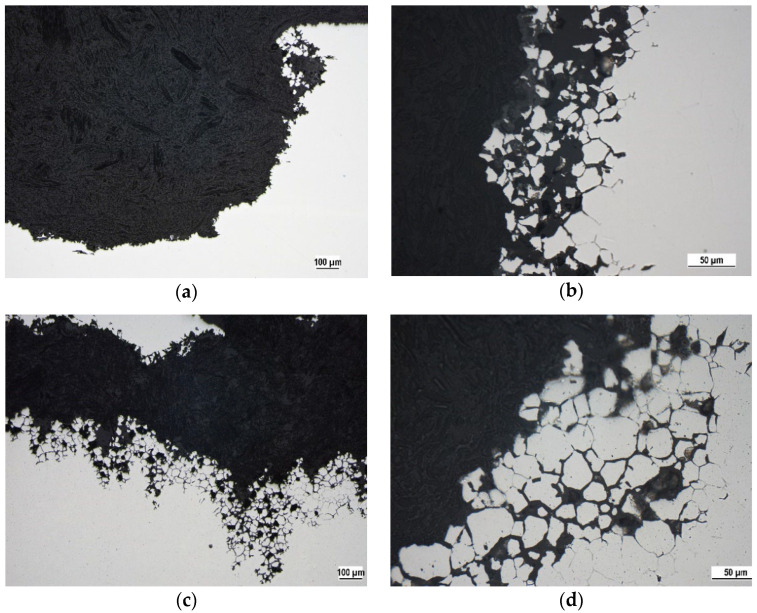
OM micrographs of the corrosion pits edges—S specimen after 24 h exposure in 1 M Cl^−^ solution (represented by 5 wt % FeCl_3_ solution), cross section: (**a**) 20 ± 3 °C, magnification 100×; (**b**) 20 ± 3 °C, magnification 400×; (**c**) 50 °C, magnification 100×; (**d**) 50 °C, magnification 400×.

**Figure 7 materials-15-08543-f007:**
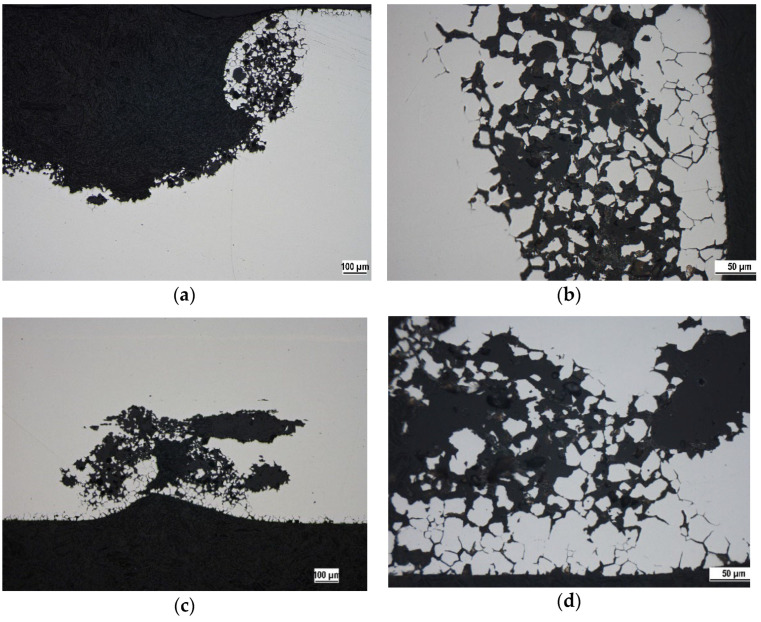
OM micrographs of the corrosion pits edges—SP specimen after 24 h exposure in 1 M Cl^−^ solution (represented by 5 wt % FeCl_3_ solution), cross section: (**a**) 20 ± 3 °C, magnification 100×; (**b**) 20 ± 3 °C, magnification 400×; (**c**) 50 °C, magnification 100×; (**d**) 50 °C, magnification 400×.

**Figure 8 materials-15-08543-f008:**
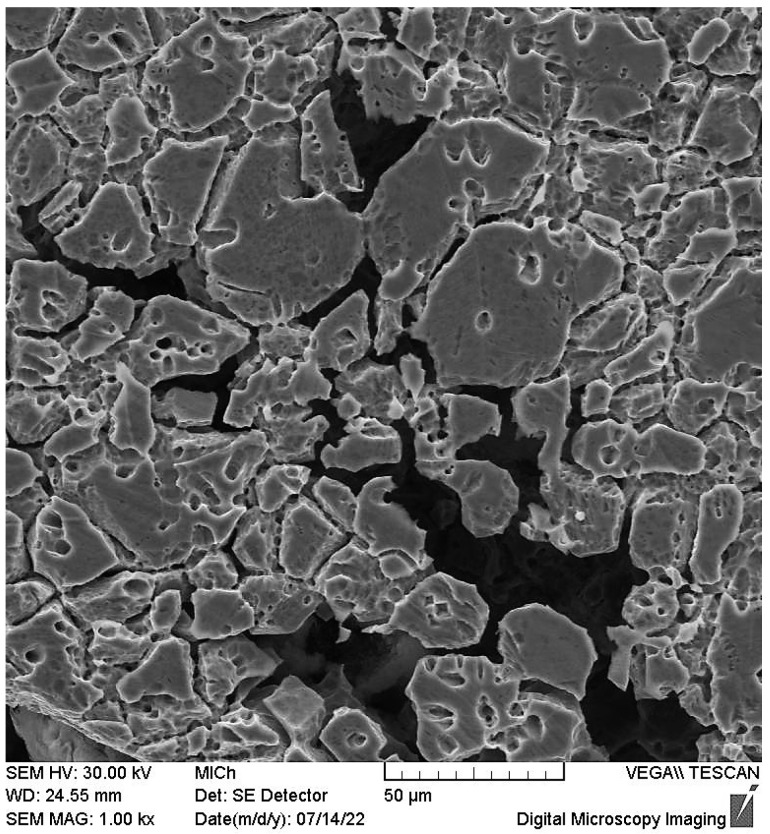
SEM micrograph—detail of the corrosion pit edge. S specimen after 24 h exposure at 20 ± 3 °C, surface area, oxalic acid electroetched (ASTM A262 practice A), 1000× magnification.

**Table 1 materials-15-08543-t001:** Chemical composition of AISI 304 SS (wt %).

Cr	Ni	Mn	N	C	Si	P	S	Fe
18.00	8.01	1.40	0.075	0.027	0.38	0.031	0.004	balance

**Table 2 materials-15-08543-t002:** Conditions of the electrochemical etching (ASTM A262, A practice).

Component	Content (wt %)	Temperature (°C)	Time (s)	Current Density (A.cm^−2^)
oxalic acid demineralized water	10 90	20 ± 3	90	1.0

**Table 3 materials-15-08543-t003:** Conditions of pickling [18].

Component	Volume (mL)	Temperature (°C)	Time (min)
HF HNO_3_ H_2_O	2 15 to 100 mL	50	10

**Table 4 materials-15-08543-t004:** Overview of tested specimen types.

Type of Surface	Specimen Designation
Sensitized (heat tinted)	S
Sensitized and pickled (without high-temperature oxides)	SP
Original non-treated	As received

**Table 5 materials-15-08543-t005:** Values of the potentiodynamic polarization parameters.

Specimen Designation and Temperature (°C)	Corrosion Potential E_corr_ (V vs. SCE)	Corrosion Current Density i_corr_ (10^−3^ mA/cm^2^)	Pitting Potential E_p_ (V vs. SCE)
As received 20	−0.23 ± 0.02	-	0.22 ± 0.01
S 20	−0.39 ± 0.03	12.1 ± 0.49	-
SP 20	−0.27 ± 0.02	-	−0.05 ± 0.04
As received 50	−0.38 ± 0.03	-	0.09 ± 0.03
S 50	−0.41 ± 0.03	24.0 ± 0.61	-
SP 50	−0.43 ± 0.03	80.5 ± 1.02	-

**Table 6 materials-15-08543-t006:** Average corrosion rates calculated from mass losses during the exposure test.

Specimen Designation and Temperature (°C)	Average Corrosion Rate (g/(m^2^ h))
As received 20	1.84 ± 0.51
S 20	12.77 ± 0.49
SP 20	14.41 ± 0.89
As received 50	15.14 ± 0.31
S 50	21.02 ± 0.37
SP 50	21.59 ± 0.38

## Data Availability

Data sharing is not applicable to this article.

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
