# Peer review of "Corrosion Behavior of Sensitized AISI 304 Stainless Steel in Acid Chloride Solution"

_materials, 2022, doi:10.3390/ma15238543_

Round 1
Reviewer 1 Report
This paper needs a major Revision based on the comments.

Author Response
"Please see the attachment."

Reviewer 2 Report
See attached file.

Author Response
"Please see the attachment."

Reviewer 3 Report
The paper is devoted to the study of corrosion resistance of sensitized austenitic stainless steel AISI 304 in chloride environment at temperatures of 20 ± 3 °C and 50 °C. The authors proposed tests on specimens pre-sensitized after heat treatment and subjected to chemical removal of high-temperature oxides to evaluate the effect of conditions on corrosion resistance. This approach succeeded in establishing corrosion resistance dependencies on pretreatment conditions and test medium.
The scientific and technical problem is to establish the dependence of corrosion resistance of austenitic stainless steel AISI 304 specimens on both pretreatment conditions (sensitization and chemical etching) and the temperature of the test medium.
The following research methods were applied: X-ray fluorescence analysis, optical microscopy and scanning electron microscopy, Potentiodynamic polarization.
The work carried out by the authors and its results definitely have scientific significance and are of great scientific interest.
I recommend the work for publication after minor corrections.
Table 5 page 8. According to the data, the table, the mass loss from the corrosion test shows a direct dependence of the loss rate on the temperature of the medium. However, between samples S20 and SP20 in the same temperature range, the difference is not critical. Should we assume that chemical removal of high-temperature oxides from the surface does not significantly affect the corrosion rate? Please ask the authors to comment. Also, the microphotographs should be zoomed in.
Author Response
"Please see the attachment."

Round 2
Reviewer 1 Report
Authors have implemented all modifications. Therefore, the manuscript can be published.
Author Response
Thank you for your review.
Reviewer 2 Report
Still in line 128, the manufacturer of PGZ100 should be added. Otherwise, it is OK after revised.
Author Response
Thank you for your review. The text was modified according to your comment (line 127)